# Serum Levels of S100B Protein and Myelin Basic Protein as a Potential Biomarkers of Recurrent Depressive Disorders

**DOI:** 10.3390/jpm13091423

**Published:** 2023-09-21

**Authors:** Lyudmila A. Levchuk, Olga V. Roschina, Ekaterina V. Mikhalitskaya, Elena V. Epimakhova, German G. Simutkin, Nikolay A. Bokhan, Svetlana A. Ivanova

**Affiliations:** 1Mental Health Research Institute, Tomsk National Research Medical Center, Russian Academy of Sciences, Tomsk 634014, Russia; roshchinaov@yandex.ru (O.V.R.); uzen63@mail.ru (E.V.M.); elenaepimakhova@mail.ru (E.V.E.); ggsimutkin@gmail.com (G.G.S.); bna909@gmail.com (N.A.B.); 2Psychiatry, Addictology and Psychotherapy Department, Siberian State Medical University, Tomsk 634050, Russia

**Keywords:** depressive disorders, depressive episode, recurrent depressive disorder, proteins of nerve tissue damage, S100B protein, myelin basic protein, glial fibrillar acid protein

## Abstract

Nowadays, nervous tissue damage proteins in serum are considered promising drug targets and biomarkers of Mood Disorders. In a cross-sectional naturalistic study, the S100B, MBP and GFAP levels in the blood serum were compared between two diagnostic groups (patients with Depressive Episode (DE, n = 28) and patients with Recurrent Depressive Disorder (RDD, n = 21)), and healthy controls (n = 25). The diagnostic value of serum markers was assessed by ROC analysis. In the DE group, we did not find changed levels of S100B, MBP and GFAP compared with controls. In the RDD group, we found decreased S100B level (*p* = 0.011) and increased MBP level (*p* = 0.015) in comparison to those in healthy controls. Provided ROC analysis indicates that MBP contributes to the development of a DE (AUC = 0.676; 95%Cl 0.525–0.826; *p* = 0.028), and S100B and MBP have a significant effect on the development of RDD (AUC = 0.732; 95%Cl 0.560–0.903; *p* = 0.013 and AUC = 0.712; 95%Cl 0.557–0.867; *p* = 0.015, correspondingly). The study of serum markers of nervous tissue damage in patients with a current DE indicates signs of disintegration of structural and functional relationships, dysfunction of gliotransmission, and impaired secretion of neurospecific proteins. Modified functions of astrocytes and oligodendrocytes are implicated in the pathophysiology of RDD.

## 1. Introduction

Depressive Disorders (DD) are a diverse group of disabling disorders that are the most common psychiatric pathology worldwide and highly recurrent [1]. Nowadays, DD is considered a multifactorial disease with various causes, such as genetic susceptibility, stress, inflammation and others. Reflecting a dysfunction of psychological and biological processes, DD manifests as clinically significant dysregulation of emotion, behavior and cognition. Knowledge of the interactions between genetic, psychosocial, immunological and neurotransmitter systems can be useful in the identification of pathogenic clues and the development of new preventive and symptomatic treatments. Structural and neurochemical violations in the prefrontal cortex accompany DD. Neurobiological processes, including neuronal damage, blood–brain barrier dysfunction, neuroplasticity, neurodegenerative processes, and the release of neurospecific proteins into the cerebrospinal fluid (CSF) and the blood, are often observed in DD. The pathological process in the central nervous system leads to astrogliosis, a marked activation of the astroglial component of the nervous tissue, resulting in the death of reactive astrocytes and disruption of the cell membrane [2]. Brain and spinal cord atrophy is caused by neurodegenerative processes, demyelination and gliosis [3]. The quantity of glial cells in the frontal areas of the cerebral cortex of patients with affective pathology is lower than in the non-psychiatric control group [4]. There is also evidence that inflammatory processes are involved in the development and progression of affective pathology [5]. The activation of microglia, the over-expression of pro-inflammatory cytokines and their receptors, and the associated chronic neuroinflammation can have a direct cytotoxic effect and lead to neurodegeneration and reduced neurogenesis [6,7]. To improve detection, treatment and prevention of recurrence of DD, it is important to understand the underlying vulnerabilities and identify specific biological markers of DD. Numerous studies suggested that DD is associated with alterations in various biological systems, such as changes in brain structure and function, immunology, endocrinology, neurotrophic factors, hormones, and oxidative stress [8]. Many authors identified the involvement of astroglial pathology in plenty of psychiatric conditions, including Depression [9,10]. Microglial cells are widely recognized as innate sentinel immune cells involved in the development of the immune and inflammatory response in the central nervous system. Microglia respond dynamically to changes in the environment, and these cells play a key role in affective immunology [11]. Damage or inflammation triggers changes in microglia morphology, motility, and expression of genes. Astrocyte is another type of glial cells that is involved in neuroinflammation by resculpting the blood–brain barrier and is essential for the normal functioning of neurons [12]. Astrocytes are also actively involved in the inflammatory reaction of nervous tissue and can alter microglial activity and neuronal function [13]. There is a unique link between microglia and astrocytes, known as astrocyte-microglia cross-talk. The astrocyte-microglia cross-talk affects emotion and affection and controls neuroinflammatory reactions [14]. Altered astrocyte–microglia cross-talk has been implicated as a primary determinant of pathological conditions and a potential therapeutic target [11]. The content of peripheral biological markers is often used to assess degeneration and inflammation in psychiatric disorders.

Nowadays, serum S100B and glial fibrillar acid protein (GFAP) are considered promising biomarkers of Mood Disorders [15,16]. S100B and GFAP are markers of astrocyte damage, reflecting the severity of the pathological process, and can be used as markers for diagnosis of brain damage [17,18]. MBP is a structural hydrophilic protein involved in maintaining the structural integrity of myelin and reflecting the degree of axon damage [19]. We suggest that the serum level of S100B, MBP, and GFAP correlates with clinical characteristics in various diagnostic groups of DD and changes after antidepressant therapy. Steinacker et al. (2021) demonstrated the possibility of using serum GFAP level identification in providing differential diagnosis and assessing the severity of DD [20]. Moreover, Schroeter et al. (2013) and Rajewska-Rager, Pawlaczyk (2016) reported perceptiveness of S100B serum levels as a marker for differential diagnosis and successful antidepressant treatment of Depressive and Bipolar affective disorders [16,21]. Disruption of myelination in Affective Disorders has also been shown in some studies [22,23]. Electron microscopy shows damaged myelin sheaths, myelin degeneration and oligodendrocyte apoptosis/necrosis in grey and white matter of the prefrontal cortex of patients with Schizophrenia and Affective Disorders. Another important task is the differential diagnosis of primary and recurrent depressive disorders (RDD) since diagnosis affects therapeutic tactics, disease prognosis, and further rehabilitation. To date, the diagnosis of Depressive Episode (DE) and RDD is based on anamnestic data indicating the presence of previous episodes of depression. At the same time, previous episodes can be deliberately dissimulated by the patient or regarded as insignificant due to the severity of the current condition. Promising are studies to find tools and methods to objectify the patient’s condition and assess the real degree of dysfunction of the nervous system.

Summing up the above, it can be assumed that astrocyte and oligodendrocyte pathology takes some part in the development of DD. To determine the specific or transdiagnostic potential of neurospecific protein markers, we compared the levels of nervous tissue damage proteins in the blood serum of patients with current DE, RDD, and healthy controls (HC).

## 2. Materials and Methods

### 2.1. Design

In a cross-sectional naturalistic study, the S100B, myelin basic protein (MBP) and GFAP levels in the blood serum were compared between two diagnostic groups (patients with DE and patients with RDD) according to ICD-10 criteria [24] and HC. The study was conducted in accordance with the Code of Ethics of the World Medical Association and approved by the Institutional Medical Review Board (Protocol of the Mental Health Research Institute local ethics committee No. 154 from 17 June 2022). All participants provided written informed consent.

### 2.2. Participants

Patients with DE (n = 28) and participants with RDD (n = 21) were recruited from the Affective States Department in the Mental Health Research Institute of the Tomsk National Research Medical Center. Inclusion criteria: a diagnosis of DD (F32, F33) according to ICD-10 [24]; ages 18–60 years. Patients with other comorbid mental disorders, for instance, Schizophrenia, Intellectual Disability, Alcoholic Psychoses, and patients with acute physical diseases, were excluded. On the first day of admission, trained psychiatrists (O.V.R., G.G.S., and N.A.B.) screened for relevant pathology for the inclusion/exclusion of subjects, disease development and the severity of the condition. The control group consisted of healthy volunteers (n = 25) recruited through local advertisements at the Mental Health Research Institute and Siberian State Medical University. A self-report questionnaire was used to screen healthy people. The questionnaire screens for physical and mental pathology, including endocrine, neurological, gynecological and psychiatric disorders.

### 2.3. Measurements

#### 2.3.1. Phenotype Measures

The severity of clinical symptoms was assessed using a set of psychometric scales: the 17-item Hamilton Depression Rating Scale (HAMD-17) for Depressive symptoms [25,26]; Hamilton Anxiety Scale (HARS) for Anxiety symptoms [27]; The Clinical Global Impression Scale (CGI) with CGI-S subscale [28] for assessing the overall clinical impression; The self-report Snaith–Hamilton Pleasure Scale (SHAPS) [29] adapted for clinical research (SHAPS-C) [30] was used to assess the severity of Anhedonia symptoms. A psychometric assessment of the patient’s condition was carried out during the first week of treatment.

#### 2.3.2. Biomarkers

Peripheral venous blood was collected from each participant between 8 and 9 a.m. after hospital admission, after eight hours of overnight fasting, and before the intake of any food or medication. Blood was collected in BD Vacutainer tubes with a clotting activator and centrifuged at 2000 rcf for 20 min at 4 °C. Serum samples were stored at −80 °C until they were analyzed.

Concentrations of calcium-binding protein (S100B), MBP and GFAP were determined in the blood serum by enzyme-linked immunosorbent assay using the DY1820-05 Human S100B DuoSet ELISA, DY4228-05 Human MBP DuoSet ELISA and DY2594-05 Human GFAP DuoSet ELISA manufactured by R&D Systems (Minneapolis, MN, USA). The reaction was set up in accordance with the attached instructions. The results of the analysis were quantified using a multimodal reader for microplate Thermo Scientific Varioskan LUX based on the Core Facility Medical genomics, Tomsk National Research Medical Center. The final results have been expressed in the units recommended by the manufacturers for the construction of calibration curves from standard portions of the analyte (pg/mL for S100B and MBP and ng/mL for GFAP).

#### 2.3.3. Statistical Analysis

Data analysis was carried out using SPSS version 26 Windows. Quantitative data in the examined sample that do not correspond to the normal distribution law (Shapiro–Wilk test) are presented as a median, lower and upper quartiles Me (Q1; Q3). When testing the null hypothesis, the critical significance level was taken at *p* = 0.05. To assess the significance of differences between groups, non-parametric Kruskal–Wallis and Mann–Whitney U tests were performed. Categorical variables were analyzed using the chi-square test. Statistically significant differences were defined as *p*-values less than 0.05. For multiple comparisons, the Bonferroni correction was used. The diagnostic value of serum markers was assessed by receiver operator characteristics (ROC) analysis. The ROC curve is plotted on the X and Y axes. It shows the frequency of true positives (sensitivity) and false positives (specificity) for each split point. 95% confidence intervals (95% CIs) were calculated for sensitivity and specificity. The area under the curve (AUC) was used to assess the informative value of the index. Correlations between protein levels and clinical and biological characteristics in the studied groups were analyzed by Spearman’s correlation test.

## 3. Results

### 3.1. Demographics and Clinical Characteristics of the Study Population

The demographics and clinical characteristics of the study population are summarized in Table 1.

In the groups of DE, RDD and HC, the average age differed (*p* = 0.196 between DE and RDD, *p* = 0.212 between DE and healthy individuals, *p* = 0.006 between RDD and healthy individuals, Mann–Whitney test). Study groups differed by sex (*p* = 0.240 between DE and RDD, *p* = 0.061 between DE and healthy individuals, *p* = 0.005 between RDD and healthy individuals, Chi-square test). In the case of RDD, the studied episode of depression was the third in the follow-up (3 (1.5; 6.5) previous episodes), and the duration of the disease was 9 (3; 17) years. The duration of the last therapeutic remission in patients with RDD was 6 (4; 18) months. Only 16 (76%) of them received maintenance therapy with selective serotonin reuptake inhibitors (SSRIs) for 10.5 (3.75; 12) months, and the rest did not receive supportive treatment or were limited to psychotherapy only. The duration of the current affective episode in the DE group was 5.5 (3; 10) months, and 3 (2; 7) months in the RDD group (*p* = 0.087, Mann–Whitney test). Most of the patients (70%, n = 34) were treated with SSRIs during hospitalization (*p* = 0.001, χ^2^). Sertraline (49%, n = 24), escitalopram (30.6%, n = 15), fluvoxamine (10.2%, n = 5), and fluoxetine (10.2%, n = 5) (*p* = 0.001, χ^2^) were used. We measured the severity of the basic clinical characteristics of patients on admission using standardized questionnaires and scales. The severity of depressive symptoms on the HAMD-17 scale is 27.5 (22.25; 32.00) in the case of DE and 31.00 (24.00; 34.00) in RDD, which corresponds to the average severity of symptoms. Anxiety symptoms, according to HARS, were 20.5 (17.00; 25.5) and 17.00 (13.00; 25.5) points, respectively, which also characterize the moderate severity. The acuteness of the anhedonia symptom measured with the SHAPS-C scale also showed no statistically significant intergroup differences. Overall, the patient’s state was assessed by clinicians as Moderately ill (4 (4; 5) and 4 (4; 4)) in the first week in both groups. The assessment of clinical characteristics of patients revealed no significant differences between the two nosological groups.

### 3.2. The Levels of S100B, MBP and GFAP in the Entire Study Group

The serum levels of S100B, MBP and GFAP as measured in the studiedgroups are shown in Table 2.

In the DE group, we did not find changed levels of S100B, MBP and GFAP compared with healthy individuals. In the RDD group, we found decreased S100B level (*p* = 0.011) and increased MBP level (*p* = 0.015) in comparison to those in HC.

### 3.3. Correlations of Clinical and Biological Characteristics in the Study Groups

Spearman’s correlation analyses of levels of S-100B, MBP and GFAP and clinical characteristics in two nosological groups are summarized in Table 3. The results of the correlation analysis showed a statistically significant relationship between clinical characteristics and serum levels of S100B, MBP and GFAP in patients with RDD. Correlation between the number of depressive episodes experienced and the level of MBP and the relationship between the duration of the last therapeutic remission and the level of GFAP were found in patients with RDD (r = −0.470; *p* = 0.037; r = −0.443; *p* = 0.050, respectively). The revealed correlations indicate an inverse relationship between clinical characteristics and the level of biological markers. The correlation analysis revealed a statistically significant relationship between the severity of depressive symptoms and biological markers (S100B, MBP) in patients with DE. Correlations between the HDRS-17 score before treatment and the level of S100B and MBP before therapy were found in patients with DE (r = 0.583; *p* = 0.001; r = 0.432; *p* = 0.022, respectively). In patients with RDD, there were no significant correlations between protein levels and mean total scale scores. In other words, an increase in the serum concentration of the studied proteins-markers of neuronal disorders is associated with greater clinical severity of depressive symptoms in patients with DE, which is not detected in patients with RDD.

Summarizing the above, we can conclude that the indicators of the studied markers of neuronal homeostasis disturbance in the recurrent course of depressive disorders, in contrast to the primary DE, cease to correlate with the severity of the studied clinical signs, which may indicate deeper neuronal disturbances.

### 3.4. ROC Analysis of Diagnostic Value of Serum Markers

The influence of markers of nervous tissue damage on the development of the current DE was determined by ROC analysis. MBP exhibited satisfactory diagnostic ability in the development of DE (AUC = 0.676; 95%Cl 0.525–0.826; *p* = 0.028) (Figure 1).

ROC analysis indicates involvement of S100B and MBP in the development of RDD (AUC = 0.732; 95%Cl 0.560–0.903; *p* = 0.013 and AUC = 0.712; 95%Cl 0.557–0.867; *p* = 0.015, respectively) (Figure 2).

Thus, provided ROC analysis indicates the influence of MBP on the development of DE, while S100B and MBP contribute to the development of RDD.

## 4. Discussion

The present study investigated the levels of S100B, MBP, and GFAP in the blood serum of newly admitted patients with DE and RDD compared to the corresponding levels in healthy individuals. The serum protein levels were measured in the morning after admission in a fasting state and before the start of any medication. The study showed that MBP contributes to the development of a DE, S100B and MBP have a significant effect on the development of RDD.

### 4.1. S100B in DD

It is known that the S100B protein produced by astrocytes is recognized as a key biochemical marker of the functional activity of neuroglial cells. Buschert et al. (2013) report that elevated levels of S100B increase neuronal plasticity in response to acute environmental stimuli, while chronic stress reduces S100B levels in the hippocampus and cerebrospinal fluid [31]. S100B could represent a protective factor in the acute phase of anxiety, yet with the development of the disease, S100B decrease may be due to decompensation. Decreased S100B levels in the serum in the examined patients are probably an indicator of impaired compensatory secretion of neurospecific proteins and dysfunction of metabolic and immune processes.

Affective pathology is characterized by impaired neuroplasticity [32,33]. Being a highly organized structure and coordinating adaptive reactions, the brain tries to restore neuroplasticity by increasing S100B levels in persons with DD. However, the compensatory–restorative mechanism and neuroimmune functioning differ in patients with depression; therefore, some researchers report elevated S100B levels in patients with Affective Disorders [34,35,36]. Other authors show a decrease or no difference in the content of the S100B protein in the blood serum and in the cerebrospinal fluid of patients with Affective and Anxiety Disorders compared to mentally healthy individuals [19,37,38,39,40]. A prospective study of children of patients with bipolar disorder has shown an abnormal neuroimmune state in bipolar offspring. This reflects a general state of vulnerability to mood disorders. The aberrant neuro–immune state manifested as increased inflammatory gene expression and decreased BDNF and S100B levels [41].

### 4.2. GFAP in DD

Unfortunately, there are only a few studies that have examined the levels of GFAP in biological fluids of patients with depression. Research by Kim et al. (2018), Miguel-Hidalgo et al. (2010), Qi with colleagues (2019) and Steinacker and co-authors (2021) indicate a relationship between the level of GFAP in biological fluids and the number of dead or damaged astrocytes, as well as the severity of DD [18,20,42,43]. Michel et al. (2021) found significantly higher levels of GFAP in the CSF of patients with unipolar depression [38]. The lower levels of GFAP in postmortem tissue samples and a reduced number of astrocytes in different cerebral regions of depressed patients compared to HC were found in several studies [44,45,46]. Our study found no significant differences in the levels of GFAP in the blood serum of patients and healthy volunteers. It does not contradict with the literature data.

### 4.3. MBP in DD

Violation of neuroplasticity, glial pathology, and dysfunction of metabolic and immune processes are accompanied by impaired connectivity and myelination, secretion of MBP by oligodendrocytes, a marker of neurodestruction [47]. Studies by Williams et al. (2019) and Valdes-Tovar et al. (2022) showed a violation of myelination processes in affective disorders [22,23]. Patients with bipolar affective disorder are characterized by decreased oligodendrocyte density in the grey and white matter of the prefrontal cortex and fronto–limbic network [48]. Studying the levels of markers MBP, S100B, and H-FABP in cerebrospinal fluid, Jakobsson et al. (2014) did not find any statistical differences between patients with bipolar disorder and HC [19]. Analysis by Tang et al. (1992) of neuropsychological characteristics of patients with chronic cerebrovascular insufficiency showed that patients suffering from DD have varying degrees of cognitive impairment and a significant increase in serum C-reactive protein, S100B, MBP and NSE [49]. Zhang et al. (2021) suggested that increased MBP gene expression in the dorsolateral prefrontal cortex of patients with Depression leads to increased plasma MBP levels [50]. We have previously found increased levels of serum MBP in patients with current DE and patients with a comorbidity of Alcohol Dependence syndrome and Affective Disorders [51]. In patients with depressive disorders, we revealed a significant increase in MBP, which is one of the main markers of destruction of the myelin sheath and indicates the processes of neurodegeneration. We also reliably confirmed the relationship between structural (increased MBP in blood serum) and functional (decreased EEG coherence) changes in the brain of patients with DD [52]. The present study shows an increase in the level of MBP in the blood serum of patients with a single DE and RDD, reaching the level of statistical significance in patients with RDD. Based on the data of J. Jakobsson et al. [19] and Zhang et al. [50], we can suggest more severe demyelination, axon damage, and neurodegeneration in patients with RDD. In addition, a significant decrease of s100B in the blood serum was shown in patients with RDD. A decreased level of s100B in the serum of patients with RDD, on the one hand, indicates the presence of chronic stress, impaired compensatory secretion of neurospecific proteins, and changes in various areas of the brain in patients with RDD. On the other hand, there is evidence that reduced concentrations of s100B have a neuroprotective effect, enhancing the proliferation of hippocampal progenitor cells and neuronal differentiation [53]. Thus, a reduced level of s100B reflects adaptive changes in neuroglia and activation of endogenous mechanisms of biological recovery and normalization of neurometabolic processes in response to damage of nervous tissue. The described above probably indicates deeper neuronal disturbances in patients with RDD, confirmed by data on more severe clinical symptoms in patients with RDD, and corresponds to the literature data.

### 4.4. Limitations

The main limitations of our study are the differences between the study groups by age and gender (*p* = 0.039 and 0.022, respectively). We performed a correlation analysis to establish a relationship between the age of patients and healthy individuals and the content of the studied proteins (Table 4). According to the analysis, no significant correlations were found between the serum S100B, MBP and GFAP levels in patients with DE and RDD and healthy individuals (*p* > 0.05).

Also, no relationship was found between the levels of S100B, MBP and GFAP in the blood serum and sex (*p* > 0.05) (Table 5).

Thus, significant correlations and interrelations between the levels of S100B, MBP and GFAP in blood serum and age and sex in the studied groups of patients with DE and RDD and healthy individuals were not revealed.

## 5. Conclusions

Despite the long duration of the disease and the number of depressive episodes in history, the clinical characteristics of DE in the case of the primary and recurrent course of an affective disorder differ slightly. At the same time, the correct differential diagnosis and verification of these disorders is very important from a therapeutic and rehabilitation point of view. The study of markers of nervous tissue damage in the blood serum of patients with a current DE indicates signs of disintegration of structural and functional relationships, dysfunction of gliotransmission, and impaired secretion of neurospecific proteins. Modified functions of astrocytes and oligodendrocytes are implicated in the pathophysiology of RDD. The question of whether the serum levels of S100B, MBP and GFAP can be used as markers for the differential diagnosis and individualized therapy of affective pathology is a relevant one.

## Figures and Tables

**Figure 1 jpm-13-01423-f001:**
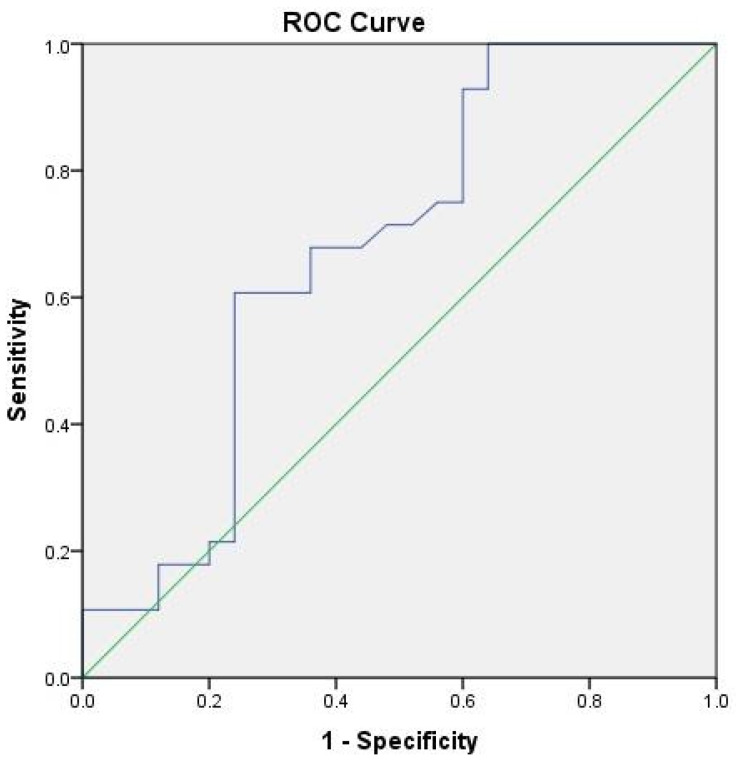
ROC Curve for prediction of depressive episode based on the serum level of MBP.

**Figure 2 jpm-13-01423-f002:**
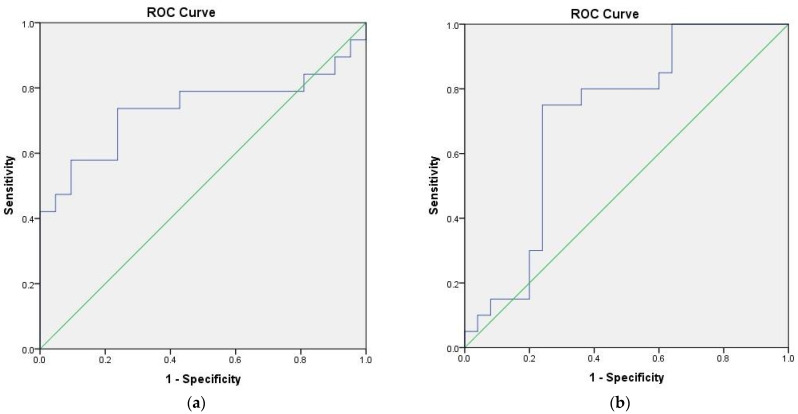
ROC Curves for prediction of recurrent depressive disorder based on the serum levels of S100B (**a**) and MBP (**b**).

**Table 1 jpm-13-01423-t001:** Demographics and clinical characteristics of the study population, Me (Q1; Q3).

Indicators	DE (n = 28)	RDD (n = 21)	HC (n = 25)	*p*-Value
Age, years	44.5 (31; 54.75)	47 (40.5; 60.5)	38 (32.5; 42.5)	**0.039 (Kruskal–Wallis)**
Gender (Male, n (%)/Female, n (%))	8 (28.57%)/20 (71.43%)	3 (14.29%)/18 (85.71%)	13 (52%)/12 (48%)	**0.022 (Chi-square test)**
Number of depressive episodes experienced (excluding current)	0	3 (1.5; 6.5)	-	
Duration of the disease (years)	1 (1; 2.5)	9 (3; 17)	-	**0.001 (Mann-Whitney)**
Duration of the last therapeutic remission (months)	-	6 (4; 18)	-	
Duration of the maintenance therapy (months)	-	10.5 (3.75; 12)	-	
Duration of the current affective episode (month)	5.5 (3; 10)	3 (2; 7)	-	*p* = 0.087 (Mann–Whitney test).
**Severity of Symptoms**				***p*-Value (Mann–Whitney test)**
HAMD-17	27.5 (22.25; 32.00)	31.00 (24.00; 34.00)	-	0.412
HARS	20.5 (17.00; 25.5)	17.00 (13.00; 25.5)	-	0.248
SHAPS	28.00 (23.00; 38.00)	32.5 (22.50; 39.00)	-	0.712
CGI-S	4.00 (4.00; 5.00)	4.00 (4.00; 4.00)	-	0.226

In bold: significant difference, *p*-value < 0.05.

**Table 2 jpm-13-01423-t002:** The serum S100B, MBP, and GFAP levels in studied population, Me (Q1; Q3).

Indicators	DE (n = 28)	RDD (n = 21)	HC (n = 25)	*p*-Value Kruskal-Wallis
S100B, pg/mL	28.76 (22.3; 44.19)	25.49 (16.86; 32.32) ***p* = 0.011 ***	33.88 (28.46; 36.56)	**0.05**
MBP, pg/mL	38.48 (28.54; 45.8)	42.17 (33; 49.09) ***p* = 0.015 ***	29.75 (18.7; 40.49)	**0.026**
GFAP, ng/mL	0.31 (0.06; 1.36)	0.12 (0.08; 1.32)	0.38 (0.13; 1.07)	0.537

In bold within group columns: * significant *p*-value of comparison with healthy control group (Mann–Whitney U-test with Bonferroni correction for multiple comparisons).

**Table 3 jpm-13-01423-t003:** Spearman’s correlations of clinical and biological characteristics in study groups.

	DE			RDD		
	S100B	MBP	GFAP	S100B	MBP	GFAP
Number of depressive episodes experienced	-	-	-	r = −0.153 *p* = 0.531	r = −0.470 *p* = 0.037	r = 0.257 *p* = 0.260
Duration of the disease	r = −0.151 *p* = 0.471	r = −0.356 *p* = 0.081	r = 0.193 *p* = 0.366	r = −0.033 *p* = 0.886	r = 0.031 *p* = 0.895	r = −0.294 *p* = 0.195
Duration of the last therapeutic remission (months)	-	-	-	r = −0.271 *p* = 0.277	r = 0.018 *p* = 0.943	r = −0.443 *p* = 0.050
Duration of the maintenance therapy (months)	-	-	-	r = 0.388 *p* = 0.112	r = 0.229 *p* = 0.345	r = 0.200 *p* = 0.398
Duration of the current affective episode (month)	r = −0.109 *p* = 0.621	r = −0.094 *p* = 0.668	r = −0.061 *p* = 0.793	r = −0.052 *p* = 0.834	r = −0.276 *p* = 0.240	r = −0.401 *p* = 0.071
HARS	r = 0.331 *p* = 0.085	r = 0.222 *p* = 0.256	r = 0.042 *p* = 0.836	r = 0.170 *p* = 0.461	r = −0.270 *p* = 0.237	r = 0.331 *p* = 0.143
HAMD−17	r = 0.583 ** *p* = 0.001	r = 0.432 * *p* = 0.022	r = 0.351 *p* = 0.072	r = −0.068 *p* = 0.769	r = −0.242 *p* = 0.290	r = 0.063 *p* = 0.786
SHAPS	r = −0.156 *p* = 0.468	r = −0.141 *p* = 0.511	r = −0.276 *p* = 0.191	r = −0.404 *p* = 0.096	r = −0.246 *p* = 0.324	r = −0.337 *p* = 0.171
CGI-S	r = 0.211 *p* = 0.280	r = 0.201 *p* = 0.304	r = 0.148 *p* = 0.463	r = 0.220 *p* = 0.337	r = −0.080 *p* = 0.730	r = 0.221 *p* = 0.337

* *p* < 0.05, ** *p* < 0.01.

**Table 4 jpm-13-01423-t004:** Spearman’s correlation matrix between protein levels and age of patients and healthy individuals.

Indicators	S100B	MBP	GFAP
DE	r = −0.080 *p* = 0.685	r = −0.193 *p* = 0.325	r = −0.234 *p* = 0.249
RDD	r = 0.425 *p* = 0.069	r = 0.313 *p* = 0.179	r = 0.055 *p* = 0.814
HC	r = 0.045 *p* = 0.847	r = −0.031 *p* = 0.882	r = −0.068 *p* = 0.751

**Table 5 jpm-13-01423-t005:** The serum S100B, MBP and GFAP levels depending on gender, Me (Q1; Q3).

Indicators	Female (n = 50)	Male (n = 24)	*p*-Value, Mann–Whitney Test
S100B	28.77 (17.36; 37.61)	31.39 (27.3; 38.36)	0.281
MBP	36.63 (25.26; 44.96)	36.93 (30.5; 50.48)	0.226
GFAP	0.55 (0.098; 1.46)	0.14 (0.07; 0.66)	0.167

## Data Availability

For ethical reasons, the datasets generated for this study will not be made available for other studies than that indicated in the protocol and informed consent form. Although not publicly available, they are available on reasonable request to Svetlana A. Ivanova (ivanovaniipz@gmail.com), following approval of the Board of Directors, in line with local guidelines and regulations.

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
