# Peer review of "Serum Levels of S100B Protein and Myelin Basic Protein as a Potential Biomarkers of Recurrent Depressive Disorders"

_jpm, 2023, doi:10.3390/jpm13091423_

Round 1

Reviewer 1 Report

This study by Levchuk et al compared the blood serum levels of S100B, MBP, and GFAP in subjects with ongoing depressive episode (DE), and recurrent depressive disorder (RDD) versus the control group. The authors found that S100B serum level was significantly decreased in subjects with RDD and MBP serum level was significantly increased in subjects with RDD, but not in subjects with DE. No significant change in GFAP was identified. The paper is clear and easy to follow, but the writing can be further improved to better help the readers understand.

Major issues:

1. In the result section 3.1, information such as the duration of the last remission in RDD subjects and the medication history of the subjects was mentioned but not shown in the table. Adding these to the table with corresponding statistics is needed. Furthermore, the authors should examine whether these factors correlate with the changes in S100B and MBP serum levels.

2. To increase the clarity of the paper:

a. Page 2, Line 53: "Mental Disorders". Using DD instead of "Mental Disorders" may be better in terms of consistency.

b. Page 2, Line 72: Introduce S100B, MBP, and GFAP briefly, including their functions; add your hypothesis of how you assume the serum level of S100B, MBP, and GFAP would change in the study.

c. The authors discussed the link of S100B, MBP, and GFAP with depression according to the literature. However, the potential mechanisms of S100B and MBP's differential changes in DE versus RDD are not discussed and should be added to the discussion section.

d. In the subtitles of the discussion section, the authors used "MD" for major depression I believe, while it was neither mentioned in previous paragraphs nor was it spelled out before. I suggest the authors use the "DD" mentioned in Line 30 for better consistency.

e. Page 8, Line 268; Spell out "CSF" for the first time of its occurrence.

Minor issues:

1. Grammar issues and typos:

a. Page 1, Line 12: "as promising drug targets";

b. Page 1, Line 24: "Altered functions of astrocytes and oligodendrocytes are";

c. Page 1, Line 40: "the liquor". Do the authors mean the cerebrospinal fluid (CSF)?

d. Page 2, Line 62: change "Astrocytes are other glial cells that are" to "The astrocyte is another glial cell type that is".

 Grammar issues and typos:

a. Page 1, Line 12: "as promising drug targets";

b. Page 1, Line 24: "Altered functions of astrocytes and oligodendrocytes are";

c. Page 1, Line 40: "the liquor". Do the authors mean the cerebrospinal fluid (CSF)?

d. Page 2, Line 62: change "Astrocytes are other glial cells that are" to "The astrocyte is another glial cell type that is".

Author Response

Dear Reviewer,

We thank you for the time spent reviewing this manuscript and for the valuable suggestions. All suggested corrections in the manuscript are highlighted in green.

Major issues:

1. In the result section 3.1, information such as the duration of the last remission in RDD subjects and the medication history of the subjects was mentioned but not shown in the table. Adding these to the table with corresponding statistics is needed. Furthermore, the authors should examine whether these factors correlate with the changes in S100B and MBP serum levels.

Response: Thank you for this recommendation. We have added appropriate information to the table 1 (please see pages 4-5). Also, we have supplemented the correlation analysis of clinical characteristics with serum levels of S100B, MBP and GFAP (please see table 3, pages 6-7).

2. To increase the clarity of the paper:

a. Page 2, Line 53: "Mental Disorders". Using DD instead of "Mental Disorders" may be better in terms of consistency.

Response: We agree with this recommendation and we have made appropriate corrections to the manuscript.

b. Page 2, Line 72: Introduce S100B, MBP, and GFAP briefly, including their functions; add your hypothesis of how you assume the serum level of S100B, MBP, and GFAP would change in the study.

Response: Thank you for this suggestion. We have added information about the functions of S100B, MBP, and GFAP and our assumptions about the levels of these markers to the introduction. Please see page 2.

c. The authors discussed the link of S100B, MBP, and GFAP with depression according to the literature. However, the potential mechanisms of S100B and MBP's differential changes in DE versus RDD are not discussed and should be added to the discussion section.

Response: Thank you for this suggestion. We have added our assumptions about the causes of different changes in S100B and MBP in patients with DE and RDD to the discussion section. Please see page 9.

d. In the subtitles of the discussion section, the authors used "MD" for major depression I believe, while it was neither mentioned in previous paragraphs nor was it spelled out before. I suggest the authors use the "DD" mentioned in Line 30 for better consistency.

Response: We have made the appropriate changes.

e. Page 8, Line 268; Spell out "CSF" for the first time of its occurrence.

Response: We changed "the liquor" (Page 1, Line 40) to “the cerebrospinal fluid (CSF)” and corrected "CSF" at Page 8, Line 268 in the manuscript.

Minor issues:

1. Grammar issues and typos:

a. Page 1, Line 12: "as promising drug targets";

Response: We have made the appropriate corrections to the manuscript.

b. Page 1, Line 24: "Altered functions of astrocytes and oligodendrocytes are";

Response: We have made the appropriate changes.

c. Page 1, Line 40: "the liquor". Do the authors mean the cerebrospinal fluid (CSF)?

Response: Yes, we meant the cerebrospinal fluid (CSF). We have made appropriate corrections to the manuscript.

d. Page 2, Line 62: change "Astrocytes are other glial cells that are" to "The astrocyte is another glial cell type that is".

Response: We agree with this recommendation and we have made the appropriate changes.

Best regards.

Reviewer 2 Report

Clinically, there is a demand to find easily measurable biomarkers that objectively indicate the patient's condition. In the presented study, the authors aimed to check if the serum level of S100B, MBP and GFAP could be a potential biomarkers of recurrence in depression.

Although there were no statistically significant differences in serum levels of postulated biomarkers between DE and RDD-groups, the authors have found some difference between RDD and healthy control groups and additionally they have performed the ROC analyses to check the utlity of proposed proteins as predictive biomarkers of DE and RDD.

I have noticed some issues which need clarification:

1) there is a significant difference in age and sex between HC and depressive patients subgroups. Some studies show changes in S100b level with age, so it was crucial in the present study to check if there was any relationship which could infuence the obtained results, and it was done here by correlation analyses. However, there was no analyses performed for antidepressant treatment history and its infulence on measured proteins. Based on the literature, e.g. S100b level may be changed by the antidepressant treatment and it even may be a predictor of antidepressant response. Was any relationship between treatment history and S100b, MBP or GFAP in the present study? Without these analysis, the results are not fully informative.

2) the measured level of GFAP was on the detection limit (or even below it) of the ELISA kit used. From the values provided in the tables 2 and 5 it also seems that there were huge inter-individual differences – how would you comment it?

3) the positive correlation between depressive symptomps severity and level of S100b only in DE group is surprising, taking into account that S100b level was not changed in DE, and in RDD it was decreased compared to control – what is the practical impact of the presented correlations?

4) there is a lot of p-values provided in the tables – it should be indicated in the text which of them were Bonferroni-corrected.

5) in Supplementary Materials there are Figures which are already provided in the manuscript.

To sum up, based on the provided data, it seems that by measuring S100b and MBP in serum it is unlikely to distinguish a first episode of depression from RDD. However, it is an interesting result among the other literature reports indicating the potential utility of these biomarkers as predictive factors in depressive disorder.

Author Response

Dear Reviewer,

The authors deeply appreciate your thorough analysis of our manuscript and valuable suggestions. All suggested corrections in the manuscript are highlighted in green.

1) there is a significant difference in age and sex between HC and depressive patients subgroups. Some studies show changes in S100b level with age, so it was crucial in the present study to check if there was any relationship which could infuence the obtained results, and it was done here by correlation analyses. However, there was no analyses performed for antidepressant treatment history and its infulence on measured proteins. Based on the literature, e.g. S100b level may be changed by the antidepressant treatment and it even may be a predictor of antidepressant response. Was any relationship between treatment history and S100b, MBP or GFAP in the present study? Without these analysis, the results are not fully informative.

Response: Thank you for this recommendation. We have supplemented the correlation analysis of clinical characteristics with serum levels of S100B, MBP and GFAP (please see table 3, pages 6-7). Blood sampling for the study was carried out on the first day after admission to the hospital, before the start of pharmacotherapy, including antidepressants. Patients with RDD at the time of admission had already completed maintenance therapy after a previous DE, if it was prescribed. This allows us to exclude the effect of drugs on the studied parameters.

2) the measured level of GFAP was on the detection limit (or even below it) of the ELISA kit used. From the values provided in the tables 2 and 5 it also seems that there were huge inter-individual differences – how would you comment it?

Response: The concentrations of GFAP were determined in the blood serum by enzyme-linked immunosorbent assay using the DY2594-05 Human GFAP DuoSet ELISA manufactured by R&D Systems (United States). This kit contains the main components for the production of enzyme immunoassay in stock concentrations. To determine the content of GFAP in blood serum, we prepared the following standards: 0.078; 0.156, 0.313; 0.625; 1.25; 2.5; 5 ng/ml. The spectrophotometer detects biotinylated and streptavidin-labelled antibodies even at concentrations below the lowest standard.

In this study, we found large inter-individual differences in GFAP content, which increase in groups of patients. This is probably due to the plasticity of the proteome, as well as the heterogeneity of the immune and inflammatory response and the role of genetic polymorphism in its development in patients with affective disorders.

3) the positive correlation between depressive symptoms severity and level of S100b only in DE group is surprising, taking into account that S100b level was not changed in DE, and in RDD it was decreased compared to control – what is the practical impact of the presented correlations?

Response: We suggest that the identification of a statistically significant relationship between clinical and biological characteristics only in the case of a primary DE may indicate deeper and more diverse homeostasis disruptions in the case of a chronic and recurrent course of disease. In other words, despite the exacerbation of differences in biomarker concentrations between RDD-patients and НС, clinical symptoms are due to the cumulative influence of various factors that have yet to be studied.

4) there is a lot of p-values provided in the tables – it should be indicated in the text which of them were Bonferroni-corrected.

Response: Thank you for this recommendation. We have indicated about Bonferroni correction in the manuscript (please see table 2, page 5-6).

5) in Supplementary Materials there are Figures which are already provided in the manuscript.

Response: Thank you for this suggestion. We have edited the Supplementary Materials.

Best regards.

Reviewer 3 Report

The authors of this MS examined the serum levels of S100B, MBP and GFAP of 234 newly admitted patients with DE and RDD, comparing them to 25 healthy controls. The explored topic is of considerable scientific and clinical interest. The paper is well within the scope of the journal and could represent an interesting contribution to the research field. The authors offer an interesting insight in the quest for biomarkers of mood diseases. 

The study design and methodology are well-explained, allowing replication and further research. The statistical methods used for data analysis are appropriate and well-described. 

The manuscript is well organized and the explored topics are presented in a comprehensive and clear manner, although authors should try to increase the quality of the figures included. Furthermore, in Table 1, authors should clarify what they mean by “Number of depressive episodes experienced” (total number of depressive episodes including or not the current one).

Before publications, the authors should check the paper for minor typos (eg. line 179: The DD-group should be corrected to DE-group).

Author Response

Dear Reviewer,

The authors deeply appreciate your thorough analysis of our manuscript and valuable suggestions. All suggested corrections in the manuscript are highlighted in green.

1. The manuscript is well organized and the explored topics are presented in a comprehensive and clear manner, although authors should try to increase the quality of the figures included. 

Response: Thank you for this recommendation. We have replaced the figures in our manuscript.

2. Furthermore, in Table 1, authors should clarify what they mean by “Number of depressive episodes experienced” (total number of depressive episodes including or not the current one).

Response: Thank you for your remark, we have clarified the table: it shows the number of depressive episodes in the anamnesis, up to the current (Number of depressive episodes experienced (excluding current)).

3. Before publications, the authors should check the paper for minor typos (eg. line 179: The DD-group should be corrected to DE-group).

Response: Thanks a lot for your suggestion. We have made the appropriate corrections to the manuscript. We will submit our article for English language editing if it is accepted.

Best regards.